**Data Availability Statement:** Data available at the following URL: https://doi.org/10.5061/dryad.q2bvq83jh

# Abnormally abrupt transitions from sleep-to-wake in Huntington's disease sheep (*Ovis aries*) are revealed by automated analysis of sleep/wake transition dynamics

William T. Schneider, Szilvia Vas, Alister U. Nicol, A. Jennifer Morton👤*

Department of Physiology, Development and Neuroscience, University of Cambridge, Cambridge, United Kingdom

* ajm41@cam.ac.uk

## Abstract

Sleep disturbance is a common and disruptive symptom of neurodegenerative diseases such as Alzheimer's and Huntington's disease (HD). In HD patients, sleep fragmentation appears at an early stage of disease, although features of the earliest sleep abnormalities in presymptomatic HD are not fully established. Here we used novel automated analysis of quantitative electroencephalography to study transitions between wake and non-rapid eye movement sleep in a sheep model of presymptomatic HD. We found that while the number of transitions between sleep and wake were similar in normal and HD sheep, the dynamics of transitions from sleep-to-wake differed markedly between genotypes. Rather than the gradual changes in EEG power that occurs during transitioning from sleep-to-wake in normal sheep, transition into wake was abrupt in HD sheep. Furthermore, transitions to wake in normal sheep were preceded by a significant reduction in slow wave power, whereas in HD sheep this prior reduction in slow wave power was far less pronounced. This suggests an impaired ability to prepare for waking in HD sheep. The abruptness of awakenings may also have potential to disrupt sleep-dependent processes if they are interrupted in an untimely and disjointed manner. We propose that not only could these abnormal dynamics of sleep transitions be useful as an early biomarker of HD, but also that our novel methodology would be useful for studying transition dynamics in other sleep disorders.

## Introduction

Sleep is essential to healthy brain function in humans. Not only does it facilitate metabolic rest [1] and support memory consolidation [2, 3], but also there is growing evidence that essential restorative and cleansing processes take place during sleep [4, 5]. Disturbed sleep can cause cognitive dysfunction and mood disturbance and exacerbate existing psychiatric problems such as depression and anxiety [6–8]. Poor sleep can also contribute to the progression of neurological diseases [9, 10] and accelerate aging [11]. Sleep problems, however, can be difficult to

**Funding:** This work was funded by a grant to AJM from the CHDI Foundation (chdifoundation.org). There is no grant number associated with the received grant. The funders had no role in study design, data collection and analysis, decision to publish, or preparation of the manuscript.

**Competing interests:** The authors have declared that no competing interests exist.

diagnose, particularly given that sleep is not a single continuum; points of transitions into and out of sleep are not always clear-cut [12, 13]. Furthermore, multiple arousals during sleep occur very frequently during the night in healthy individuals [14, 15]. Whilst such arousals may not be noticed by normal sleepers, self-reporting of awakenings is high in insomniacs and people with other health problems, such as sleep apnoea [16]. In such disorders, nocturnal awakenings can be stressful and are associated with damage to neural and physiological function [9, 17, 18].

It is thought that a minimum period of uninterrupted sleep is required for sleep-dependent mechanisms such as memory consolidation [19] and restorative processes including glymphatic cleaning [18]. In studies where sleep fragmentation is induced experimentally, typically using sounds [20, 21] or physical disturbance [22], the sleeper is awaked into a sudden hyper-aroused state. Even in the short term, these induced arousals are associated with an array of problems including increased stress, mood disorders, and cognitive and memory impairments [22–24]. Such studies typically quantify the level of sleep disturbance by recording the time spent in bed and asleep, the number of awakenings, the duration of the awakening, and resulting daytime symptoms such as daytime sleepiness [25]. However, the dynamics of sleep transitions or awakenings are rarely investigated despite it being likely that it is the disturbance to sleep rather than the amount of sleep that is lost that causes physiological stress [24]. Therefore, further study of the temporal and spectral dynamics of sleep/wake transitions in models of sleep disturbance is warranted.

A number of neurodegenerative diseases including Huntington's (HD) [26], Parkinson's [27], and Alzheimer's disease [28] are associated with progressively disturbed sleep [29]. HD patients exhibit a range of sleep problems, including increased nocturnal awakenings [30–35]. HD is a progressive, dominantly inherited neurodegenerative disease caused by an unstable CAG repeat mutation in the *HD* gene [36]. The sleep abnormalities associated with HD can appear during its presymptomatic and early stages [37]. We have previously hypothesised that poor sleep may aggravate symptoms in HD patients [30]. In light of emerging evidence from the Alzheimer's disease field that sleep is essential for clearing metabolites that may be neurotoxic, such as beta amyloid [38], it is possible that inadequate sleep will also accelerate disease progression in HD. There is also growing awareness of the importance of sleep in synaptic homeostasis [39, 40] and that sleep deficits may exacerbate the need for synaptic compensation caused by HD [41]. It is not clear to what extent abnormal sleep is part of the disease itself, or simply a consequence of anxiety and depression which are common in early-stage HD [42, 43]. Nevertheless, given that these mental health problems are associated with poor sleep, there is also potential for them to feedback and exacerbate problems with sleep in HD patients. A better understanding of sleep abnormalities in HD is clearly desirable.

Here we investigated the dynamics of spontaneous transitions between (non-rapid eye movement) NREM sleep and wake in normal and HD transgenic sheep (OVT-73) [44]. The HD sheep were at an early/presymptomatic stage, showing no signs of disease. Behaviourally, they appear normal during both wake and sleep. They do, however, show abnormalities in their pattern of sleep and electroencephalograms (EEG) power distribution across the night [45]. We analysed EEG recorded for 10 hours during a single undisturbed night from these sheep. We then used an automated algorithm developed in-house to quantify dynamics of transitions between sleep and wake. We found no difference between HD and normal sheep in general measures of sleep, including the length of NREM sleep periods and the number of awakenings. There were, however, clear differences between HD and normal sheep in the dynamics of awakenings. In particular, sleep-to-wake transitions were both faster and of greater magnitude in HD than they were in normal sheep. By contrast, there were no genotypic differences in the speed and severity of transitions into sleep. These data suggest that even though

there are no behavioural signs of disease and little brain pathology at this age, control of the sleep/wake cycle in HD sheep is already abnormal. If similar events are present in human HD sufferers then they may experience a decline in the rejuvenating processes of sleep even before any signs of the disease are present. Our automated method for investigating the dynamics of sleep transitions provides a new tool for quantifying a previously unappreciated component of sleep that has potential as an early biomarker in HD.

## Methods

### Subjects

Ten female merino sheep aged 6 years were used in this study. Five were transgenic and carried a full-length copy of the human *HD* gene with a CAG repeat of 73 [44], and the other (controls) 5 were age-matched flock-mates. These sheep were part of a longitudinal comparative study of EEG in HD and normal sheep conducted at the Pre-clinical Imaging Research Laboratory (PIRL), Adelaide, South Australia. Both sleep [45] and drug studies [46] published recently were part of this longitudinal study. Note that the sleep recordings analysed in this study were made before the sheep were treated with any psychoactive drugs. Our study was approved by the SAHMRI Animal Ethics Committee and animal handling followed the physical containment conditions (PC2) set by the Institutional Biosafety Committee and the Office of the Gene Technology Regulator (OGTR, Australia).

The sheep were raised on open pasture as part of a larger flock at a breeding facility. Candidate sheep selected for electrophysiological recordings were transported to PIRL where implantation surgery was conducted. The number of animals selected was determined as the optimal number that could be effectively managed at the facility under PC2 conditions whilst providing a balanced sample of transgenic and normal genotype subjects. Following surgery, the sheep were held in individual pens in a covered outdoor arena where they were exposed to natural daylight conditions. The pens were lined with clear Perspex so that the sheep were in visual contact with others in neighbouring pens and at the same time the risk of damage to the implanted devices was minimized. The sheep were provided with food and water twice daily, at the start and end of the working day. Animal care technicians monitored the animals daily for physical condition and food and water consumption. Veterinary care was always available on site.

### Surgery

EEG electrodes were implanted surgically 1 year before the recordings were made, as described previously [45–47]. Briefly, implants consisted of eight subdural recording electrodes and one subdural reference electrode to record EEG. The eight EEG electrodes were positioned bilaterally at locations corresponding to the postcruciate gyrus, the ansatus sulcus, the front third of the ectolateral sulcus, and the lateral sulcus near the anterior part of the entolateral sulcus. Electrodes were also positioned above the eyes for recording eye movements (EOG), and stainless steel coils were implanted in the neck muscles for recording electromyogram (EMG).

### Recording

EEG recordings were made continuously from an hour before sunset to an hour after sunrise (10.6 hrs) from unrestrained sheep behaving naturally. EEG was recorded using a wireless telemetry system (Advanced W2100-System, Multichannel Systems Gmbh, Germany). Data were saved at a sampling rate of 250 Hz. EEG data were re-referenced using a global reference prior to importing into MATLAB for sleep transition analyses. Data from all sheep were

recorded indoors on the same night. Sheep had been moved to the recording pens on the previous day. Each sheep was randomly allocated to an individual pen in sight of other sheep within the room and had free access to food and water. Pens were located in two rooms containing six (3 HD, 3 normal) or four (2 HD, 2 normal) pens each. Lights were left on during the night allowing sleep scoring to be validated against video footage. The data and MATLAB code used in this article are available online (https://doi.org/10.5061/dryad.q2bvq83jh).

## Automated sleep/wake transition classification

We developed a novel, automated method to detect transitions in-and-out of sleep from the EEG. Briefly, the power spectral density for each 10 second epoch were obtained using a Fast Fourier Transform (FFT). Epochs were removed as artefacts if their spectral power exceeded three times the median absolute deviation from the mean spectral power of all epochs. Classification of sleep state was based on state space ratio 2 (SSR2, Fig 1A) using a modification of the state space analysis method [48] that we adapted previously for use with sheep [49]. In this method, epochs were automatically classified as either sleep, wake, or inter-state based upon relative thresholding (Fig 1B). A distance of 20% (the value '$y$' in Fig 1B), from the mean location of the vigilance stages was chosen empirically as it provided a conservative threshold between states and ensured a wide separation between them that contained a transitional state. Only transitions between NREM sleep and wake were analysed. Using the manually-scored data, any transitions that contained epochs scored as rapid eye movement (REM) sleep were removed from the analysis. Transitions were confirmed *post hoc* by cross checking with manually-scored NREM sleep and wake epochs (manually scored as previously described [49]). In

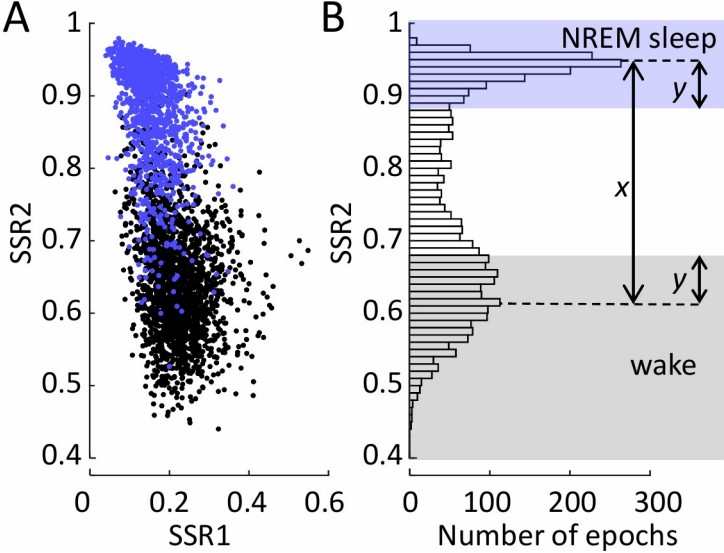

**Fig 1. Method for automated classification of vigilance state using state space ratios. (A)** State space ratios (SSRs) of each wake (black symbols) and non-rapid eye movement (NREM) sleep (blue symbols) epoch during a single night, for an example (normal) sheep. The two frequency ratios used are SSR1 (6.5–9/0.5–9 Hz) and SSR2 (0.5–20/0.5–100 Hz). **(B)** Histogram of all epochs during the recording night according to their SSR2 value, for the same sheep. The histogram is overlaid with an indicative diagram of the automated scoring method. The value $x$ is the distance between the two highest peaks of the histogram, and length $y$ is 20% of $x$. All epochs with SSR2 values between 1 and the value at the upper peak minus $y$, are classified as NREM sleep (highlighted by blue shading). All epochs with SSR2 values below the value at the lower peak plus $y$, are classified as wake (highlighted by grey shading). As the distribution of SSR2 values differs between sheep and channel, so do the $x$ and $y$ values and peak locations that define the NREM sleep and wake boundaries. This method ensures that these boundaries are relative to each sheep/channel, and importantly, that they cannot overlap.

our automated method, a transition from sleep-to-wake was defined as the time when more than 50% of epochs across all channels, within a rolling 5-minute window, met the automatic classification of wake. The transition centre was defined as the mid-point on the slope of the transition between the two states. The same method was used to find the transitions from wake-to-sleep. The length of bouts of wake or NREM sleep were calculated as the time difference between the start of wake, to the start of NREM sleep, or vice versa. Data from all sheep were sufficiently noise-free to conduct these analyses.

For all sheep, spectral powers of epochs during transitions were normalised to the total power in the entire night (using the whole spectrum; 0 to 125 Hz). Normalisation was performed for each channel of each sheep separately. Speed of transitions was assessed by three methods as follows: (A) By using the value of the peak in the moving variance (in a 1-minute window) across all frequencies (normalised at each frequency by total power in each frequency bin) from -2 to +2 minutes from the transition centre (Fig 2); (B) By calculating the time difference from when the EEG power exceeded or dropped below more than the standard deviation around the mean of the power during the 5 minutes prior to the transition, to when EEG power returned to within the standard deviation of the power in the 5 minutes after the transition and (C) By calculating gradient in power change using a straight line from the start and end points of the transition as found in method B. Where analyses were conducted with the spectrum separated into frequency bands, the ranges we used were as follows: delta = 0.5–4 Hz; theta = 4–9 Hz; alpha = 9–14 Hz; beta = 14–35 Hz; and gamma = 35–125 Hz [46]. EEG powers in the 50 Hz frequency bin were omitted due to an artefact caused by a cycle in mains power. Spectral analysis of pre/post transition differences was performed from -4 to -2 mins before transition, and from +2 to +4 mins after transition, because during these windows transitions had either not yet begun, or had already finished. During all stages of data processing investigators were blind to sheep genotype.

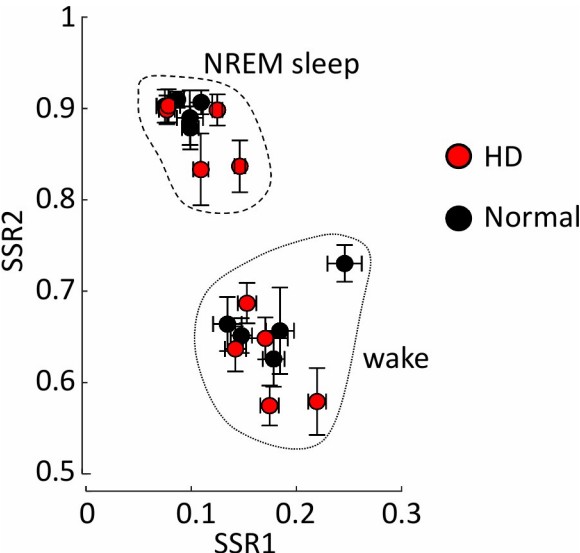

**Fig 2. Comparison of sleep states in normal and Huntington's disease sheep.** Scatter plot showing the mean state space ratio locations for all epochs in each sheep (N = 5 for each genotype) separated by the automated scoring method. NREM sleep and wake epochs are enclosed by dotted lines to show the separation. There are no differences in the locations of the mean state space ratios for NREM sleep and wake between HD (filled circles) and normal (empty circles) sheep. Data shown are means taken across all channels, error bars are standard deviation. Each sheep is represented by an individual symbol. Note that some symbols overlap.

## Statistical analyses

Statistics were performed in R version 4.0.0. Two sample t-tests were used to test for differences between genotypes (N = 5; HD, N = 5; normal) in general measures of sleep fragmentation. Differences in wake/sleep transition times between HD and normal sheep across all channels were analysed using generalised linear mixed models with Gamma distributions. Single animals were considered as the experimental unit, though data from all eight channels for each sheep were included in the analyses, thus sheep ID was included as a random variable. Transition gradient and variance differences were analysed using linear models, again with sheep ID included as a random variable and data were normalised for the linear models using sqrt() transform. For transitions to wake, where all gradients were negative, gradient and variance data were normalised first using abs() transform. Pre- and post-transition power data were also analysed using linear mixed models, with sheep ID included as a random variable.

## Results

### General features of the sleep wake cycle are not abnormal in HD sheep

There were no differences between HD and normal sheep in measures of sleep that are typically used to assess sleep fragmentation, including number of transitions, total power, and lengths of NREM sleep and wake bouts (Table 1). State space analysis of the automatically classified wake and NREM sleep vigilance stages also showed no differences in the mean spectral ratios of vigilance stages between HD and normal sheep (Fig 2).

### Transition speed of sleep-to-wake but not wake-to-sleep differs between genotypes

Spectral analysis (Fig 3) showed that the speed of transition from sleep-to-wake was slower in normal (Fig 3B) than it was in HD sheep (Fig 3D). This was confirmed by all three of the methods we used for quantifying transition speed. First, while the peak moving variance across all frequencies was similar in normal (Fig 3E) and HD (Fig 3G) sheep for wake-to-sleep transitions, it was higher during the transition from sleep-to-wake in HD (Fig 3H) than in normal sheep (Fig 3F). Second, duration of sleep-to-wake transitions were shorter (approximately half as long) in HD sheep in the delta, theta, and gamma frequency ranges (Table 2). Third, the gradients of the sleep-to-wake transitions were steeper in HD than in normal sheep in the delta, theta, and alpha ranges (Table 2). In contrast to this, there were no differences between HD and normal sheep in any of these measures of speed for the wake-to-sleep transitions (Table 3).

**Table 1. General measures of sleep fragmentation.**

| | Sleep parameters | | | |
| | Genotype | | Statistics | |
| Measure | Normal | Transgenic | p | t (df) |
|---|---|---|---|---|
| Number of transitions to wake | 11.6 (± 2.2) | 10.4 (± 1.9) | 0.692 | -0.41 (8) |
| Number of transitions to sleep | 18.4 (± 3.1) | 17.2 (± 1.5) | 0.738 | -0.35 (8) |
| Mean length of wake episode (mins) | 14.5 (± 3.0) | 17.2 (± 2.4) | 0.437 | 0.82 (8) |
| Mean length of sleep episode (mins) | 10.5 (± 1.2) | 9.5 (± 0.5) | 0.478 | -0.74 (8) |

Data presented are mean values ± standard error of the mean and statistical values from t-tests.

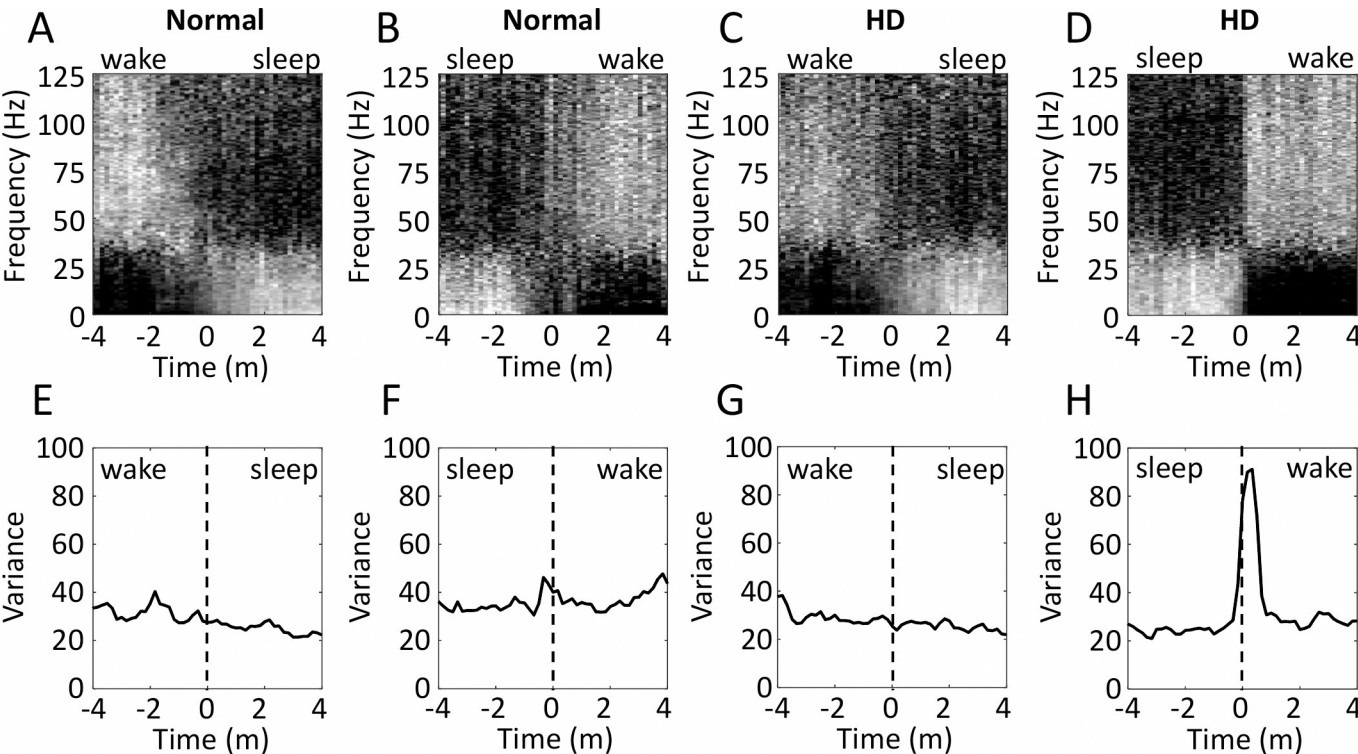

**Fig 3. Transitions from sleep-to-wake are more abrupt in Huntington's disease than normal sheep.** Panels A-D show spectrograms of mean transitions from wake-to-sleep (**A, C**) or from sleep-to-wake (**B, D**), for normal (**A, B,** N = 5) and HD (**C, D,** N = 5) sheep. Each frequency bin in the spectrograms is normalised to itself, such that black represents the minimum power at that frequency during a transition, and white is the maximum. Graphs in **E-H** show the variance in these normalised frequency power values across all frequencies during a 1 minute moving window. Data in this figure are shown from the A2 L channel.

## Spectral analysis shows sleep-to-wake dynamics differ between HD and normal sheep

During transitions-to-wake, there were associated changes to dynamics of EEG power spectra. The magnitude of the changes depended upon whether the power was considered as a proportion of its average during wake (Fig 4A, 4C, 4E and 4G) or sleep (Fig 4B, 4D, 4F and 4H). Calculating proportions relative to both the average during wake and sleep in this way is useful

**Table 2. Differences in durations and gradients of transitions to wake for each frequency band between normal and HD transgenic sheep.**

| Frequency band | Durations of transitions to wake (mins) | | | | Gradients of transitions to wake (power/mins) | | | |
|---|---|---|---|---|---|---|---|---|
| | Genotype | | Model output | | Genotype | | Model output | |
| | Normal | Transgenic | p | t | Normal | Transgenic | p | t |
| Delta | 2.06 ± 0.06 | 1.07 ± 0.04 *** | < 0.001 | -3.72 | -0.208 ± 0.018 | -0.572 ± 0.040 ** | 0.006 | -3.73 |
| Theta | 1.94 ± 0.08 | 1.09 ± 0.06 ** | 0.001 | -3.25 | -0.064 ± 0.007 | -0.200 ± 0.017 ** | 0.008 | -3.53 |
| Alpha | 1.85 ± 0.10 | 1.33 ± 0.12 | 0.083 | -1.73 | -0.013 ± 0.001 | -0.045 ± 0.004 ** | 0.006 | -3.73 |
| Beta | 1.45 ± 0.12 | 1.23 ± 0.12 | 0.311 | -1.01 | -0.009 ± 0.001 | -0.032 ± 0.005 * | 0.032 | -2.60 |
| Gamma | 1.83 ± 0.10 | 1.04 ± 0.07 ** | 0.003 | -2.95 | 0.010 ± 0.001 | 0.020 ± 0.002 | 0.086 | -1.96 |

Data presented are mean values ± standard error of the mean. Significance codes:

*p < 0.05,

**p < 0.01,

*** p < 0.001.

**Table 3. Differences in durations and gradients of transitions to sleep for each frequency band between normal and HD transgenic sheep.**

| | Durations of transitions to sleep (mins) | | | | Gradients of transitions to sleep (power/mins) | | | |
|---|---|---|---|---|---|---|---|---|
| | Genotype | | Statistics | | Genotype | | Statistics | |
| Frequency band | Normal | Transgenic | p | t | Normal | Transgenic | p | t |
| Delta | 1.74 ± 0.06 | 2.05 ± 0.11 | 0.603 | 0.52 | 0.390 ± 0.024 | 0.273 ± 0.023 | 0.105 | 1.83 |
| Theta | 1.69 ± 0.08 | 1.73 ± 0.13 | 0.998 | 0.00 | 0.092 ± 0.004 | 0.069 ± 0.018 | 0.967 | -0.04 |
| Alpha | 1.73 ± 0.08 | 1.78 ± 0.12 | 0.985 | 0.02 | 0.019 ± 0.001 | 0.025 ± 0.002 | 0.513 | -0.684 |
| Beta | 1.88 ± 0.11 | 1.71 ± 0.12 | 0.624 | -0.49 | 0.009 ± 0.001 | 0.014 ± 0.002 | 0.173 | -1.45 |
| Gamma | 1.51 ± 0.08 | 1.55 ± 0.18 | 0.774 | 0.29 | -0.009 ± 0.001 | -0.009 ± 0.001 | 0.881 | -0.16 |

Data presented are mean values ± standard error of the mean. Significance codes:

*$p < 0.05$, **$p < 0.01$, *** $p < 0.001$.

because the balance of powers in the EEG spectra differs between genotypes in the two states, thus the scale of the transitional change of powers also differs between genotypes depending on whether it is considered as changes relative to average power in sleep or wake. Generally, however, the patterns were broadly similar. Two minutes or so prior to waking, power in the delta (Fig 4A and 4B), alpha (Fig 4C and 4D), and beta (Fig 4E and 4F) ranges were lower than the average during sleep, and in the minute preceding the awakening power in these ranges fell rapidly towards levels that are normal during wake. This was true for both normal and HD sheep. However, the level of delta power prior to waking was lower in normal than in HD sheep (p = 0.016, t = -3.03; Fig 4A); in normal sheep, prior to waking, delta power was ~ 50% lower than its average level during sleep, while in HD sheep the reduction in power before waking was much smaller (~25% less; Fig 4A). Thus, the scale of the eventual transition in delta power is smaller for normal sheep than it is for HD sheep. We found similar differences in transition dynamics in the alpha (p = 0.009, t = -3.41; Fig 4D) and beta (p = 0.003, t = -4.32; Fig 4F) frequency ranges proportional to average wake power. In the gamma range there was no difference between HD and normal sheep prior to the transition, but gamma was lower in HD compared to normal sheep after waking (p = 0.013, t = 2.56; Fig 4G), relative to average NREM power.

### Transition from wake-to-sleep dynamics in the context of average sleep and wake power are similar in HD and normal sheep

The only significant difference between genotypes in the dynamics of EEG power was in the alpha range where it increased more in HD than it did in normal sheep after falling asleep (relative to average wake power; p = 0.012, t = -2.57; Fig 5D). Otherwise, transitions into sleep were very similar in HD and normal sheep in terms of spectral power dynamics and speed of transition (see for example, delta and alpha power in Fig 5A, 5B and 5C; data from beta and theta bands are not shown).

### Discussion

We used a novel automated analysis of quantitative EEG to examine the dynamics of sleep/wake transitions in normal and HD sheep. We found that transitions from sleep-to-wake in HD sheep are faster than those of normal sheep. Furthermore, these awakenings amount to a more extreme change in EEG powers in HD than in normal sheep. By contrast, transitions from wake-to-sleep were not abnormal in HD sheep. We did not note any overt differences in behaviour between HD and normal sheep [45], suggesting that the rapid awakenings are not

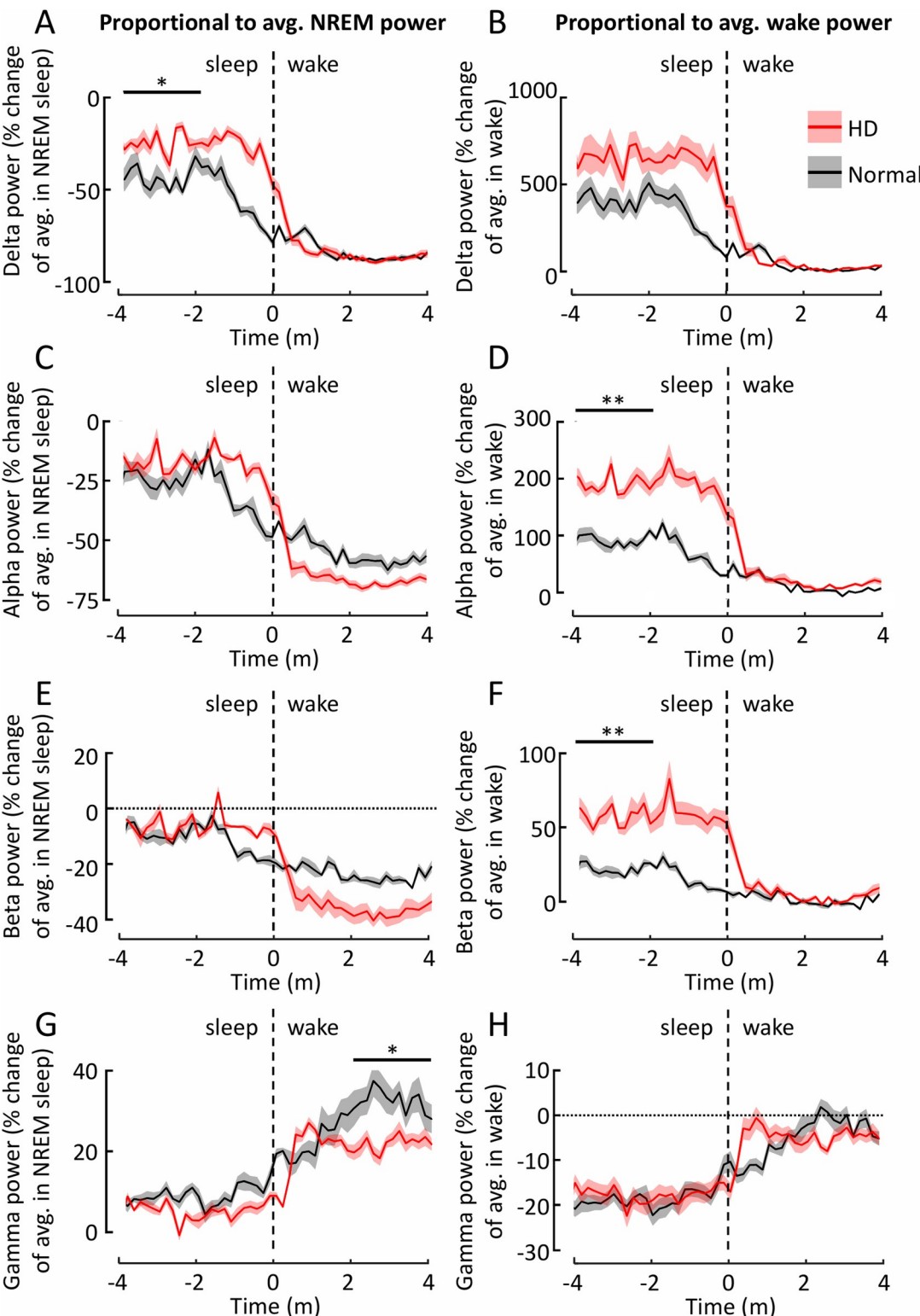

**Fig 4. Changes in power during transitions from sleep-to-wake are more extreme in Huntington's disease than normal sheep.** Power changes were calculated as the proportional change relative to the mean NREM sleep power (**A, C, E, G**) or wake power (**B, D, F, H**). Frequency ranges presented are delta (**A, B**), alpha (**C, D**), beta (**E, F**) and gamma (**G, H**). Each trace represents proportional power changes for a particular frequency band in the mean transitions to wake for all normal (black line) or HD (red line) sheep. The shading represents the standard error of the mean. For clarity data are shown for the Fr2-L

channel only. Statistics reported were performed with data from all channels included. avg. = average, * = P < 0.05, ** = P < 0.01.

manifest as an overtly abnormal behaviour. A focused monitoring of behaviour would be needed to confirm this. Nevertheless, given that in symptomatic HD patients and animal models of HD, sleep becomes fragmented as the disease progresses [30, 31, 50] and that the sheep we used were at a very early stage of disease, we suggest that the sudden awakenings we see in the HD sheep are among the first of the sleep disorders that occur in this HD model.

Our study raises new questions about both the process underlying normal awakenings and the mechanism underlying early changes in awakening in HD. In the normal sheep, transitions from sleep-to-wake showed a characteristic pattern, with a gradual decline in delta, theta, and alpha powers that started around two minutes before awakening. Even prior to this decline, slow wave power was nearly half its normal level during NREM sleep, making the eventual transition into wake a 'gentle' event. Smooth transitions between sleep and wakefulness is regulated by a balanced interaction of excitatory and inhibitory networks of subcortical and cortical brain structures [51, 52]. Abnormal function of either network may modify the dynamics

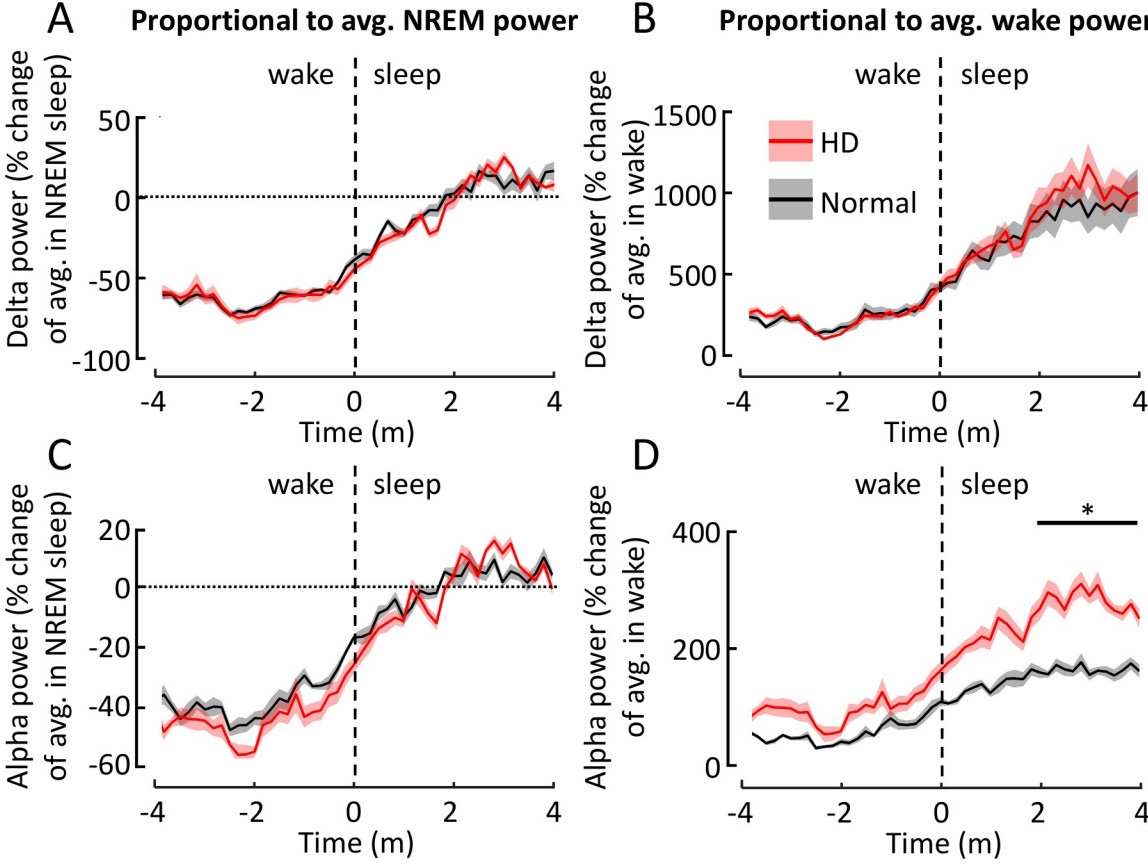

**Fig 5. Similar changes in power during transitions from wake-to-sleep are seen in normal and Huntington's disease sheep.** Power changes across transitions from wake-to-sleep were calculated as the proportional change relative to the mean NREM sleep power (**A, C**) or wake power (**B, D**). Each trace represents proportional power changes for a particular frequency band in the mean transitions to sleep for all normal (black line) or HD (red line) sheep. Frequency ranges presented are delta (**A, B**) and alpha (**C, D**). The shading represents the standard error of the mean. For clarity data are shown only for the Fr2-L channel only. Statistics reported were performed with data from all channels included. avg. = average, * = P < 0.05, ** = P < 0.0.

of these transitions. Of particular relevance, loss of hypothalamic orexinergic neurons causes severe disruption to the sleep/wake cycle, with low thresholds for transitions, as is seen in narcoleptic patients [53] and orexin knockout mice [54]. Several studies have reported loss of orexinergic neurons in post mortem brains of HD patients [55–57] and HD mice [58]. Orexinergic system damage may therefore contribute to the abnormal awakenings that we see in HD sheep. Furthermore, altered function of the histaminergic system, which is closely linked to the orexinergic system in regulating the sleep/wake cycle [59, 60] may be another potential cause of abnormal sleep in HD patients [61, 62]. Treatments targeting the histaminergic system (using antagonists on the histamine 3 [H3] receptor) have been suggested specifically for treating early stage HD [63]. Given that the abnormalities we found were present in presymptomatic HD sheep, there may be potential in investigating therapies such as H3 antagonists that might target sleep/wake abnormalities in this model of HD.

It is notable that whereas sleep-to-wake transitions are abnormal, there was little effect on wake-to-sleep transitions in the HD sheep. While at face value this suggests that the control of the arousal system may be affected first in HD, it is important to remember that wake and sleep are generated by different pathways which, although interconnected, are modulated by different factors [52]. Furthermore, HD is a multifactorial disease with known impairments in multiple neurotransmitter systems [58, 64, 65]. Thus, there are multiple possible explanations for our findings. For example, if NREM sleep-promoting pathways (such as the melanin-concentrating hormone) that inhibit wake-promoting pathways are impaired early in HD, then this may result in abnormal transition dynamics without a change in the length of the sleep episode. Alternatively, given that both neuronal function and connectivity are impaired in HD [66, 67] it is possible that the ability of the cortical neurons to produce and propagate slow waves [68] is altered in HD, making it easier for them to quickly transition from the hyperpolarised sleep-like state (cortical OFF state) to a depolarised wake-like state (cortical ON state).

The functional relevance of the gentle awakening seen in normal sheep that is impaired in HD sheep is unknown. We can speculate that in normal sleepers, there is a period of transition to wake that is required in order for sleep-related processes to be terminated. Examples of such processes include sleep spindle-associated memory consolidation, which is a vital component in the memory-creating process that occurs during NREM sleep [69] and reduces in activity prior to waking [70]. Slow wave activity (delta power) is also considered key to much of the restorative function in sleep [71], including memory consolidation [72, 73]. Slow wave activity is highest when sleep pressure is elevated and also reduces during the progression of sleep [52, 74]. Thus, the lack of a reduction in slow wave power prior to awakening, as we have observed in HD sheep, may indicate that these important functions are not progressing as normal during sleep. Although this is speculation on our part it is supported by considerable evidence showing impairment of memory in patients with disorders that cause sudden awakenings, such as sleep apnoea [75], insomnia [76] and Alzheimer's disease [77, 78]. Memory consolidation in HD sheep has not been studied to date but would be a worthwhile future study.

Cognitive function is impaired in HD patients and correlated with changes in sleep [33]. There is little doubt that improving sleep would have a beneficial knock-on effect on cognitive function in HD patients. It may be that the optimal time for starting such treatments is when the first deficits appear, rather than when sleep structure is abnormal enough to affect the behaviour of the patient. In this study we have identified a very early sleep abnormality in HD sheep using a novel automated method. This technique can be applied easily to EEG recorded from HD patients and we were able to obtain results from only a single night of recording. If established, such a biomarker may be a relatively undemanding method for tracking the progression of HD. This could be especially useful if it were possible to perform the assessment

using only a small number of sleep/wake transitions during the day. Further work is clearly needed to understand sudden awakenings and their consequences for HD patients.

## Supporting information

**S1 Checklist. The ARRIVE guidelines 2.0: Author checklist.**
(PDF)

## Acknowledgments

We are grateful to the staff at The Preclinical Imaging & Research Laboratory, SAHMRI, Gillies Plains, Adelaide, South Australia for their excellent assistance in gathering the data used in this study, and the late Professor Tim Kuchel for his unwavering support of the project.

## Author Contributions

**Conceptualization:** William T. Schneider.

**Data curation:** Alister U. Nicol.

**Formal analysis:** William T. Schneider, Szilvia Vas.

**Funding acquisition:** A. Jennifer Morton.

**Investigation:** William T. Schneider.

**Methodology:** William T. Schneider.

**Project administration:** A. Jennifer Morton.

**Software:** William T. Schneider.

**Supervision:** A. Jennifer Morton.

**Validation:** William T. Schneider.

**Visualization:** William T. Schneider.

**Writing – original draft:** William T. Schneider.

**Writing – review & editing:** William T. Schneider, Szilvia Vas, Alister U. Nicol, A. Jennifer Morton.

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
