## [Decision Letter · Decision Letter 0]

21 Apr 2021

PONE-D-21-08690

Abnormally abrupt transitions from sleep-to-wake in Huntington’s disease sheep (Ovis aries) are revealed by automated analysis of sleep/wake transition dynamics

PLOS ONE

Dear Dr. Morton,

Thank you for submitting your manuscript to PLOS ONE. After careful consideration, we feel that it has merit but does not fully meet PLOS ONE’s publication criteria as it currently stands. Therefore, we invite you to submit a revised version of the manuscript that addresses the points raised during the review process.

We look forward to receiving your revised manuscript.

Kind regards,

Vladyslav Vyazovskiy, PhD

Academic Editor

PLOS ONE

Journal Requirements:

3. Thank you for including your ethics statement: 

"Our study adhered to the requirements of the SAHMRI Animal Ethics Committee and animal handling followed the physical containment conditions set by the Institutional Biosafety Committee and the Office of the Gene Technology Regulator (OGTR, Australia).".   

a. Please amend your current ethics statement to confirm that your named ethics committee specifically approved this study.

For additional information about PLOS ONE submissions requirements for ethics oversight of animal work, please refer to http://journals.plos.org/plosone/s/submission-guidelines#loc-animal-research  

4. PLOS ONE has specific criteria regarding the reporting of animal research (https://journals.plos.org/plosone/s/submission-guidelines#loc-animal-research).

Specifically, these guidelines require that details regarding care and monitoring are clearly stated. To that effect, please include the following in your Methods section:

- A description of the animals' housing conditions, including food/water, temperature, and lighting conditions

- How animals were monitored for health and behavior throughout the study, and the timeline of the study protocol

5. As part of your revision, please complete and submit a copy of the ARRIVE Guidelines checklist, a document that aims to improve experimental reporting and reproducibility of animal studies for purposes of post-publication data analysis and reproducibility:

https://arriveguidelines.org/sites/arrive/files/Author%20Checklist%20-%20Full.pdf

Please include your completed checklist as a Supporting Information file.

Note that if your paper is accepted for publication, this checklist will be published as part of your article.

7. We note that you have stated that you will provide repository information for your data at acceptance. Should your manuscript be accepted for publication, we will hold it until you provide the relevant accession numbers or DOIs necessary to access your data. If you wish to make changes to your Data Availability statement, please describe these changes in your cover letter and we will update your Data Availability statement to reflect the information you provide.

Reviewers' comments:

Reviewer's Responses to Questions

**Comments to the Author**

1. Is the manuscript technically sound, and do the data support the conclusions?

Reviewer #1: Yes

Reviewer #2: Yes

2. Has the statistical analysis been performed appropriately and rigorously? 

Reviewer #1: Yes

Reviewer #2: Yes

3. Have the authors made all data underlying the findings in their manuscript fully available?

Reviewer #1: Yes

Reviewer #2: Yes

4. Is the manuscript presented in an intelligible fashion and written in standard English?

Reviewer #1: Yes

Reviewer #2: Yes

5. Review Comments to the Author

Reviewer #1: This study describes the physiological transition from sleep to wake and wake to sleep in a sheep model of HD. The paper is very well written, the methods described fully, the figures support the reported results, and the discussion is an excellent balance of caution with appropriate speculation. I have no suggestions for changes to this manuscript.

Reviewer #2: This is a well written study examining sleep dynamics in sheep genetically engineered with Huntington's disease. The authors find that HD sheep transition from NREM sleep to wakefulness more rapidly than normal sheep. I only have one major concern. The authors report that 8 EEG electrodes were implanted in the animals. However, it is unclear where the electrodes were placed and which were used for the analyses. Although the authors reference an earlier paper for the exact surgical methods, I think they should be restated in this paper. Also, I would like to know how the data from the 8 electrodes were handled. Were all channels examined independently of the others? Were transitions from NREM to wake more rapid in all channels (brain regions) in HD sheep? Was the synchrony of state transitions across channels different between HD and normal sheep? Were there hemispheric differences in synchrony?

I would also be interested to know if there were any behavioral differences in how HD and normal sheep awaken. For example, did HD sheep awaken with an abrupt startle response? Did environmental factors (e.g sound, other sheep) precipitate awakenings?

6. PLOS authors have the option to publish the peer review history of their article (what does this mean?). If published, this will include your full peer review and any attached files.

Reviewer #1: No

Reviewer #2: No

---

## [Author Response · Author response to Decision Letter 0]

30 Apr 2021

Dear Vlad, 

Thank you for the reviewer’s and your helpful comments on the manuscript. We have changed the manuscript according to these points, and addressed them in order below (our responses are in red text). We have also now uploaded our data and source code to a data repository (link will become live upon publication - https://doi.org/10.5061/dryad.q2bvq83jh). This link has been added to the updated manuscript. 

I look forward to seeing this paper in press. Again, thanks for your input,

Best wishes, 

Jenny Morton

Journal Requirements:

Figure filenames have been corrected.

Formatting of manuscript has been corrected to reflect style requirements. 

The references have been double-checked. We are not aware of any retracted papers in our citations. We have added a recently accepted paper (Vas et al, 2021) that describes sleep and EEG changes in the HD and normal sheep.

3. Thank you for including your ethics statement: 

"Our study adhered to the requirements of the SAHMRI Animal Ethics Committee and animal handling followed the physical containment conditions set by the Institutional Biosafety Committee and the Office of the Gene Technology Regulator (OGTR, Australia).". 

a. Please amend your current ethics statement to confirm that your named ethics committee specifically approved this study.

The statement has been amended to:

"Our study was approved by the SAHMRI Animal Ethics Committee and animal handling followed the physical containment conditions set by the Institutional Biosafety Committee and the Office of the Gene Technology Regulator (OGTR, Australia)." 

The updated statement is in the submission form. 

For additional information about PLOS ONE submissions requirements for ethics oversight of animal work, please refer to http://journals.plos.org/plosone/s/submission-guidelines#loc-animal-research

4. PLOS ONE has specific criteria regarding the reporting of animal research (https://journals.plos.org/plosone/s/submission-guidelines#loc-animal-research).

Specifically, these guidelines require that details regarding care and monitoring are clearly stated. To that effect, please include the following in your Methods section:

- A description of the animals' housing conditions, including food/water, temperature, and lighting conditions

- How animals were monitored for health and behavior throughout the study, and the timeline of the study protocol

We have now included a paragraph in the ‘Subjects’ section of the Methods in which this information is detailed. 

5. As part of your revision, please complete and submit a copy of the ARRIVE Guidelines checklist, a document that aims to improve experimental reporting and reproducibility of animal studies for purposes of post-publication data analysis and reproducibility:

https://arriveguidelines.org/sites/arrive/files/Author%20Checklist%20-%20Full.pdf

Please include your completed checklist as a Supporting Information file.

Note that if your paper is accepted for publication, this checklist will be published as part of your article.

This has been included as a Supporting Information file. 

The reference to these data has now been removed. 

7. We note that you have stated that you will provide repository information for your data at acceptance. Should your manuscript be accepted for publication, we will hold it until you provide the relevant accession numbers or DOIs necessary to access your data. If you wish to make changes to your Data Availability statement, please describe these changes in your cover letter and we will update your Data Availability statement to reflect the information you provide.

URL for data (will be live at publication): https://doi.org/10.5061/dryad.q2bvq83jh

URL for review (will download all files in .zip)

https://datadryad.org/stash/share/STwHOi_Qwe6idFHA34ehQvyYHwuEX589dW_hnakLC00

Reviewers' comments:

Reviewer's Responses to Questions

Comments to the Author

1. Is the manuscript technically sound, and do the data support the conclusions?

Reviewer #1: Yes

Reviewer #2: Yes

2. Has the statistical analysis been performed appropriately and rigorously?

Reviewer #1: Yes

Reviewer #2: Yes

3. Have the authors made all data underlying the findings in their manuscript fully available?

Reviewer #1: Yes

Reviewer #2: Yes

4. Is the manuscript presented in an intelligible fashion and written in standard English?

Reviewer #1: Yes

Reviewer #2: Yes

5. Review Comments to the Author

Reviewer #1: This study describes the physiological transition from sleep to wake and wake to sleep in a sheep model of HD. The paper is very well written, the methods described fully, the figures support the reported results, and the discussion is an excellent balance of caution with appropriate speculation. I have no suggestions for changes to this manuscript.

Thank you for your very positive feedback. 

Reviewer #2: This is a well written study examining sleep dynamics in sheep genetically engineered with Huntington's disease. The authors find that HD sheep transition from NREM sleep to wakefulness more rapidly than normal sheep. I only have one major concern. The authors report that 8 EEG electrodes were implanted in the animals. However, it is unclear where the electrodes were placed and which were used for the analyses. Although the authors reference an earlier paper for the exact surgical methods, I think they should be restated in this paper. Also, I would like to know how the data from the 8 electrodes were handled. Were all channels examined independently of the others? Were transitions from NREM to wake more rapid in all channels (brain regions) in HD sheep? Was the synchrony of state transitions across channels different between HD and normal sheep? Were there hemispheric differences in synchrony?

Thank you for your very helpful comments. The methods have now been published in a second paper, so a full description of the surgery has not been added. In response to the reviewer’s question, reference to this recent paper, as well as a brief description of the surgical methods have been added (Line 111).

Channels were not analysed independently, instead all eight channels were used in the analyses and included as repeated measures. Wording in the statistical methods has been updated to make this clearer (line 198). We did not see consistent differences between channels in the speed of transition into wake. There were also no obvious differences between HD and normal sheep in the synchrony of state transitions or any hemispheric differences. It may be that such differences in channels exist, though a larger sample size would be required to detect them. 

I would also be interested to know if there were any behavioral differences in how HD and normal sheep awaken. For example, did HD sheep awaken with an abrupt startle response? Did environmental factors (e.g sound, other sheep) precipitate awakenings?

We did not observe any behavioural changes that suggested a change in behaviour upon awakening. Although animals were videoed for observation during the study, this was done using a low frequency video that was not suitable for detailed behavioural assessments of sleep awakening. Nevertheless, given our findings, this would be very interesting to assess in detail in future studies. A comment to this effect has been added to the manuscript (Line 312).

In the interval since this study was submitted for publication, an in-depth qEEG assessment conducted on these sheep has been published (ref 45), that includes some behavioural analysis. While it does not address specifically the manner in which the sheep awaken (for reasons explained above), the behavioural data are consistent with abnormalities in awakenings described in the current manuscript. A comment about those behavioural findings and reference to this work has been added to the manuscript (Lines 312 - 318). 

6. PLOS authors have the option to publish the peer review history of their article (what does this mean?). If published, this will include your full peer review and any attached files.

Do you want your identity to be public for this peer review? For information about this choice, including consent withdrawal, please see our Privacy Policy.

Reviewer #1: No

Reviewer #2: No

---

## [Editor Report · Decision Letter 1]

3 May 2021

Abnormally abrupt transitions from sleep-to-wake in Huntington’s disease sheep (Ovis aries) are revealed by automated analysis of sleep/wake transition dynamics

PONE-D-21-08690R1

Dear Dr. Morton,

We’re pleased to inform you that your manuscript has been judged scientifically suitable for publication and will be formally accepted for publication once it meets all outstanding technical requirements.

Kind regards,

Vladyslav Vyazovskiy, PhD

Academic Editor

PLOS ONE

---

## [Editor Report · Acceptance letter]

5 May 2021

PONE-D-21-08690R1 

Abnormally abrupt transitions from sleep-to-wake in Huntington’s disease sheep (*Ovis aries*) are revealed by automated analysis of sleep/wake transition dynamics 

Dear Dr. Morton:

I'm pleased to inform you that your manuscript has been deemed suitable for publication in PLOS ONE. Congratulations! Your manuscript is now with our production department. 

Kind regards, 

on behalf of

Dr. Vladyslav Vyazovskiy 

Academic Editor

PLOS ONE